# Phenotypic and Genetic Heterogeneity of Adult Patients with Hereditary Spastic Paraplegia from Serbia

**DOI:** 10.3390/cells11182804

**Published:** 2022-09-08

**Authors:** Stojan Perić, Vladana Marković, Ayşe Candayan, Els De Vriendt, Nikola Momčilović, Andrija Savić, Nataša Dragašević-Mišković, Marina Svetel, Zorica Stević, Ivo Božović, Šarlota Mesaroš, Jelena Drulović, Ivana Basta, Igor Petrović, Olivera Tamaš, Milija Mijajlović, Ivana Novaković, Dragoslav Sokić, Albena Jordanova

**Affiliations:** 1Neurology Clinic, University Clinical Center of Serbia, Faculty of Medicine, University of Belgrade, 11000 Belgrade, Serbia; 2VIB-UAntwerp Center for Molecular Neurology, VIB, 2610 Antwerpen, Belgium; 3Department of Biomedical Sciences, University of Antwerp, 2610 Antwerpen, Belgium; 4Neurosurgery Clinic, University Clinical Center of Serbia, Faculty of Medicine, University of Belgrade, 11000 Belgrade, Serbia; 5Molecular Medicine Center, Department of Medical Chemistry and Biochemistry, Medical University-Sofia, 1431 Sofia, Bulgaria

**Keywords:** hereditary spastic paraplegia, genetic testing, pure HSP, complicated HSP

## Abstract

Hereditary spastic paraplegia (HSP) is among the most genetically diverse of all monogenic diseases. The aim was to analyze the genetic causes of HSP among adult Serbian patients. The study comprised 74 patients from 65 families clinically diagnosed with HSP during a nine-year prospective period. A panel of thirteen genes was analyzed: L1CAM (SPG1), PLP1 (SPG2), ATL1 (SPG3A), SPAST (SPG4), CYP7B1 (SPG5A), SPG7 (SPG7), KIF5A (SPG10), SPG11 (SPG11), ZYFVE26 (SPG15), REEP1 (SPG31), ATP13A2 (SPG78), DYNC1H1, and BICD2 using a next generation sequencing-based technique. A copy number variation (CNV) test for SPAST, SPG7, and SPG11 was also performed. Twenty-three patients from 19 families (29.2%) had conclusive genetic findings, including 75.0% of families with autosomal dominant and 25.0% with autosomal recessive inheritance, and 15.7% of sporadic cases. Twelve families had mutations in the SPAST gene, usually with a pure HSP phenotype. Three sporadic patients had conclusive findings in the SPG11 gene. Two unrelated patients carried a homozygous pathogenic mutation c.233T>A (p.L78*) in SPG7 that is a founder Roma mutation. One patient had a heterozygous de novo variant in the KIF5A gene, and one had a compound heterozygous mutation in the ZYFVE26 gene. The combined genetic yield of our gene panel and CNV analysis for HSP was around 30%. Our findings broaden the knowledge on the genetic epidemiology of HSP, with implications for molecular diagnostics in this region.

## 1. Introduction

Hereditary spastic paraplegia (HSP) is a clinically heterogeneous disease caused by axonal degeneration of the thoraco-lumbar portion of the corticospinal tract and of the medullary and cervical portions of the dorsal columns, while some pathological changes may also be seen in the spinocerebellar tract. HSP is a rare disorder with a global average prevalence of 1.8 per 100,000 individuals [1]. It has an onset from infancy to late adulthood, and the disease course is usually slowly progressive.

Patients may present with a pure HSP phenotype with spastic lower limb weakness, frequent urinary urgency, and mildly impaired vibration sensibility, or with complicated HSP phenotypes where additional features are seen, such as thin corpus callosum, intellectual disability, dementia, brain white matter hyperintensity lesions (WMHLs), cerebellar atrophy and ataxia, polyneuropathy, muscle atrophy, movement disorders, macular degeneration, optical atrophy, cataracts, hearing loss, etc. [2]. In line with this, Agosta and colleagues [3] used advanced MRI techniques to show microstructural brain abnormalities in extra-motor regions, suggesting a more diffuse degenerative process in a heterogeneous group of HSP patients.

Besides phenotypic heterogeneity, HSP is among the most genetically diverse of all monogenic diseases. It can be inherited in autosomal dominant, autosomal recessive, X-linked or rarely in a maternal pattern. More than 80 genes have been reported to be associated with HSP [4,5]. Development of next generation sequencing (NGS) technology identified many new genes responsible for HSP, but still for more than 50% of the cases, causative genes are not identified [6,7].

The aim of this study was to perform a systematic genetic analysis of a representative cohort of adult Serbian HSP patients using a gene panel approach. Notably, there have not been any clinical or genetic data on HSP patients from Serbia or other Western Balkan countries so far. The Neurology Clinic at the University of Belgrade is the largest neurological center in Serbia and several surrounding countries. The clinic follows up the majority of patients with rare neurological disorders in the region. Approximately two thirds of all HSP patients clinically diagnosed in Serbia are referred to this center. Thus, analysis of our HSP cohort may have important implications for genetic epidemiology and the genetic work-up of HSP patients from Serbia and neighboring countries.

## 2. Patients and Method

This study comprised all adult patients clinically diagnosed with HSP at the Inpatient and Outpatient Units of the Neurology Clinic, Clinical Center of Serbia, during a nine-year prospective period from 1 January 2009 until 31 December 2017. The following diagnostic criteria for HSP were applied: 1. symptomatic, approximately symmetric spastic weakness of both lower extremities with hyperreflexia and usually with extensor plantar response; 2. presence or absence of urinary urgency or incontinence; 3. exclusion of all other diseases that may mimic HSP. Detailed information about patients’ family members was gathered through a structured interview. All symptomatic family members that were alive and willing to participate in the genetic analysis were tested. Unfortunately, we were not able to perform trio analysis in most of the sporadic cases since our patients were adults and their parents were not usually alive. However, when a variant of unknown significance (VUS) was found, we included all available family members in the segregation analysis.

In all tested patients, the following diseases were excluded based on disease history, clinical findings, additional investigations, and genetic analysis where necessary: amyotrophic lateral sclerosis, primary lateral sclerosis, spinocerebellar ataxia, dopa responsive dystonia, myelitis, HIV, syphilis, multiple sclerosis, cerebral palsy, structural brain and spinal cord damage, adrenoleukodystrophy, adrenomyeloneuropathy, and B12 deficiency. Blood samples were collected from all patients and sent to the Center for Molecular Neurology at the University of Antwerp for genetic analysis. The study was approved by the Ethical Board of the Neurology Clinic, Clinical Center of Serbia and all patients signed informed consent to participate in the study.

Clinical data were collected from the electronic medical records, including: gender, age at disease onset, age at examination, walking ability, presence of hyperreflexia, weakness, muscle atrophy, sensory abnormalities and sphincter disturbances, and analysis of HSP plus features (skin anomalies, short stature, scoliosis, foot deformities, adducted thumbs, abnormal saccades, nystagmus, optic atrophy, cataract, deafness, dysarthria, dysphagia, cognitive impairment, epilepsy, ataxia, tremor, dystonia, and other extrapyramidal signs). Magnetic resonance imaging (MRI) of the brain and/or of the spinal cord was performed in the majority of patients. Where possible, nerve conduction studies (NCS) and somatosensory evoked potentials (SSEP) were also assessed.

We have analyzed the following 13 HSP genes: *L1CAM* (SPG1), *PLP1* (SPG2), *ATL1* (SPG3A), *SPAST* (SPG4), *CYP7B1* (SPG5A), *SPG7* (SPG7), *KIF5A* (SPG10), *SPG11* (SPG11), *ZYFVE26* (SPG15), *REEP1* (SPG31), *ATP13A2* (SPG78), *DYNC1H1*, and *BICD2* using an in-house developed next generation sequencing-based multiplex amplification of specific targets for the resequencing (MASTR) method. Our gene panel was designed in 2017 and included genes that were reportedly common at the time. Using the MASTR technology, we obtained 93% coverage over all 288 amplicons. The remaining 7% of amplicons were covered by Sanger sequencing. All identified genetic variants were confirmed by two independent Sanger sequencing reactions. The results were annotated following the recommendations of the Human Genome Variation Society (HGVS). The copy number variation (CNV) test for *SPAST* (encoding protein *spastin*), *SPG7* (encoding protein *paraplegin*) and *SPG11* (encoding protein *spatacsin*) was also performed using MLPA analysis. The interpretation of pathogenicity for all variants were done according to the American College of Medical Genetic and Genomics (ACMG) guidelines. Methods of descriptive statistics were used. All data obtained are presented in the manuscript.

Haplotype analysis was performed in the Roma patients carrying the p.L78* mutation, as well as in two patients from the P1 and P2 Italian families, where p.L78* was initially reported [8]. Samples from the Italian patients were provided by courtesy of Dr. M. Bassi, Scientific Institute E. Medea, Italy. Markers in the p.L78* mutation region described by Arnoldi et al. were used for the haplotype analysis. STR markers were amplified with fluorescently labelled primers. Subsequently, the PCR products were electrophoretically separated on an ABI3730xl DNA Analyzer, using GeneScanTM 500 Liz Size Standard (Applied Biosystems). Genotyping results were analyzed by the Local Genotype Viewer software (https://www.neuromicssupportfacility.be/, accessed on 6 September 2022). The SNPs were genotyped by Sanger sequencing.

## 3. Results

During a nine-year period, 74 patients (48.6% females) from 65 families followed up at the Neurology Clinic at the University of Belgrade fulfilled the inclusion criteria. The mean age at testing was 41.9 ± 12.9 years. In 4.1% of the cases, the disease started in infancy, 10.8% in childhood, 17.6% in adolescence, 39.2% in early adulthood (between the ages of 21 and 40), and 28.4% in late adulthood. Among our HSP patients, 12 (18.5%) had a clear autosomal dominant pattern of inheritance, while five additional patients from five (7.7%) families likely had autosomal dominant inheritance. Eight (12.3%) families had autosomal recessive inheritance pattern. One patient had a probable X-linked inheritance, while in one patient, the inheritance pattern was ambiguous since she was adopted. The remaining patients were sporadic cases.

After gene panel and CNV analyses, conclusive genetic findings were obtained in 19 (29.2%) families. In this way, genetic diagnosis could be established in nine (75.0%) of 12 families with clear autosomal dominant inheritance and two (25.0%) of eight with autosomal recessive inheritance (Figure 1). Detailed genetic findings are listed in Table 1.

Sixteen HSP patients from 12 (18.5%) of 65 families had mutations in the *SPAST* gene, causing SPG4 (Table 2 and Appendix A). Almost all of them had a positive family history congruent with the autosomal dominant inheritance. The disease usually started in early adulthood, although there were patients with onset in the first year of life and even after 60 years of age. The course was insidious and none of the patients used walking aids at the time of examination. Almost half of the patients complained of sphincter disturbances. The majority of our SPG4 patients had a pure HSP phenotype. Two siblings (HSP9 and HSP10 from Family 2) had dysarthria as an additional symptom with no other clinical or radiological signs of cerebellar impairment and with no other features of complicated HSP. It is of note that patient HSP12, son of HSP11 from Family 3, had epilepsy due to right hippocampal sclerosis. Interestingly, in family 10, where two siblings and their mother were affected, significant intrafamilial clinical variability was noted. The sister, HSP80, had disease onset in her thirties. At the age of 43, she was ambulant, having spasticity, hyperreflexia, and weakness more pronounced in her lower limbs, sphincter disturbances, and decreased vibration sensibility in her lower limbs. Additionally, mild cerebellar ataxia and several non-specific white matter hyperintensity lesions (WMHLs) were noted. The affected brother, HSP81, had symptoms from birth including lower limb spasticity, walking difficulties, foot deformity, and deafness with consequent speech problems. He was still ambulant at the age of 33.

Three (4.6%) families had conclusive findings in the *SPG11* gene coding protein *spatacsin* (Table 3), with all cases being sporadic. All identified *SPG11* mutations were recurrent and therefore further segregation analysis was not performed. In all patients, the disease started in adulthood, and gait was abnormal, but none of them used aids. Two *SPG11* patients had a complex phenotype. Patient HSP35 had mild intellectual disability (IQ 78 with MMSE 29), mild dysarthria, and mild tremor with thin corpus callosum and with periventricular WMHLs. Patient HSP113 had foot dystonia. She is an engineer, successfully working at the age of 57, but cognitive testing was not performed. Patient HSP128 also had no cognitive complaints, but, unfortunately, a detailed cognitive evaluation could not be performed.

Two sporadic patients from two apparently unrelated (3.1%) Roma families carried the c.233T>A (p.L78*) mutation in a homozygous state in the *SPG7* gene encoding *paraplegin* protein. Notably, in the extended family of HSP119, there were several affected individuals in consecutive generations; however, the history of consanguineous marriages and the presence of the recessive p.L78* allele in the proband suggested a pseudo-dominant inheritance of HSP in the pedigree. Familial analysis did not identify additional affected members in the extended pedigree of HSP18. P.L78* was initially described as a founder mutation in two Italian families by Arnoldi et al. [8]. We performed haplotype sharing analysis in HSP118 and HSP119 and compared them with the haplotypes of the Italian patients reported to carry the same mutation. We observed a shared p.L78* haplotype between all cases (Table 4).

Patient HSP119 had the disease onset in late adulthood but was able to walk without any aid at the age of 57 (Table 5). He had additional features of cervical dystonia and severe cortical and cerebellar atrophies on MRI without clinical signs of cognitive impairment or ataxia. Similarly, Patient HSP118 had the disease onset in late adulthood. At the age of 54 he used a unilateral support to walk. The only additional feature in this patient was a postural hand tremor.

Patient HSP34 had a heterozygous, likely pathogenic c.746T>C (p.Leu249Pro) variant in the *KIF5A* gene, causing SPG10 (Table 6). The variant was not present in the parents or other members of the family, suggesting de novo mutation in this case. Her disease started during childhood, and the patient had complicated HSP with axonal polyneuropathy and mild postural hand tremor. In the sporadic patient DIST690, we identified two compound heterozygous mutations in the *ZFYVE26* gene causing SPG15: one mutation (c.2114dupC; p.E706*) was present in the patient and her asymptomatic father, while another mutation (c.2357del; p.P786Hfs*10) was present in the patient and her asymptomatic mother, confirming segregation in the family. The disease onset was during the teenage years of the patient, and the phenotype was complicated with foot dystonia and mild cerebellar signs (dysarthria, limb ataxia, and tremors). She had no ocular impairments, audiometry was normal, as well as nerve conduction studies (NCS) and EEG. Neuropsychological testing showed normal results.

## 4. Discussion

As in other populations, 18% of our HSP families and 63% of genetically diagnosed patients had Strumpell–Lorrain disease, i.e., SPG4 due to mutations in the *SPAST* gene [9]. The Italian HSP network analyzed more than 1700 HSP patients and found that mutations in the *SPAST* gene were the most frequent single cause of HSP, comprising 38% of familial cases and 19% of sporadic cases [10]. This percentage increases to up to 80–90% in Chinese and Taiwanese autosomal dominant HSP patients [11,12]. The majority of our SPG4 patients had a pure HSP phenotype starting in early adulthood, although two patients had an earlier disease onset. This form of HSP has been reported to have disease onset from the age of 1 until 80, with a median onset in the fourth decade of life [13,14,15,16]. In our SPG4 patients, the disease had a slowly progressive course and none of them used walking aids at the time of examination, which is in line with the literature data that these patients are able to walk for several decades after the disease onset [17]. Almost 50% of our SPG4 patients had vibration sensibility impairment and sphincter disturbance. Column sensory deficit was variably presented in SPG4 patients in previous studies, while bladder disturbance was present in approximately one third of the cases [14,15]. Two siblings from our SPG4 cohort had dysarthria with no other signs of cerebellar impairment. Mild spastic dysarthria can be considered as a part of the pure SPG4 phenotype [16]. SPG4 usually presents as pure HSP, but rarely, complex phenotypes are seen, including cerebellar ataxia, executive dysfunction, epilepsy, psychosis, WMHLs, arachnoid cyst of the posterior fossa, polyneuropathy, hand tremors, and amyotrophy of the hand muscles [10,15,18,19,20,21]. In accordance with this, modern methods of MRI revealed a widespread affection of gray (and secondarily white) matter in SPG4 cases, including corpus callosum, medio-dorsal thalami, parieto-occipital regions, upper brainstem, and cerebellum, suggesting at least subclinical involvement of structures other than the pyramidal tract [22].

Around 5% of our HSP families, i.e., 16% of genetically confirmed cases, had mutations in the *SPG11* gene coding protein *spatacsin*. SPG11 is the most common form of autosomal recessive HSP, comprising 20–30% of these patients [10,23]. The disease usually starts in adolescence, and by the age of 25, most patients develop full clinical manifestations. However, our patients had adult-onset disease and a similar observation was described by other authors [24,25]. Interestingly, two Czech SPG11 patients with late-onset disease were compound heterozygotes for the c.5381T>C variant that was also described in our patients with late-adult onset [26]. One of our SPG11 patients had pure HSP, and two had a complex phenotype with symptoms such as cognitive problems, dysgenesis of corpus callosum, periventricular WMHLs, dysarthria, tremors, and dystonia. SPG11 typically presents with a complex phenotype comprising different signs: thin corpus callosum, intellectual disability, progressive cognitive decline, cerebellar ataxia, WMHLs (“ears of the lynx”), seizures, parkinsonism, tremors, macular lesions, polyneuropathy, amyotrophy, etc. [27,28,29,30]. The disease course is worse than in autosomal dominant HSP with wheelchair dependance after 15 years of disease duration. However, none of our patients, the majority of them being in their sixth decade of life, used assistance while walking. SPG11 is the most common form (60%) of HSP associated with a thin corpus callosum; conversely, atrophy of the corpus callosum can be seen in 40–80% of SPG11 patients [17,31].

We found two patients from two (3%) HSP families with *SPG7* variants. SPG7 accounts for up to 10% of all HSP patients globally [10]. *SPG7* mutations can cause both pure and complex phenotypes, especially later in the course of the disease, including cerebellar ataxia and atrophy, optic atrophy, chronic progressive external ophthalmoparesis, supranuclear palsy, cognitive impairment, thin corpus callosum, and polyneuropathy [10,16,32,33]. It is of note that in our two Romani patients we identified a biallelic c.233T>A (p.L78*) mutation in the *SPG7* gene. This mutation was initially reported as a founder allele in an Italian cohort by Arnoldi et al. [8]. It was also found in French patients, both in compound heterozygous and homozygous states [32]. In a Dutch cohort of 60 SPG7 patients, one patient presenting with HSP and ataxia syndrome was a compound heterozygous for p.Leu78* with another *SPG7* mutation [34]. One Romani patient in a Spanish cohort was found to carry the nonsense p.Leu78* mutation in a heterozygous state, while his mother and aunt were homozygous for this mutation. Large rearrangements were excluded in this patient, but a second mutation was not found, though the authors could not rule out that their patient had a second mutation in non-analyzed SPG7 regions (e.g., the promoter region) [35]. Another homozygous patient for the p.Leu78* mutation was described in Italy with late-adult onset pure HSP [36]. Besides the pure HSP phenotype, the biallelic p.Leu78* mutation was also associated with progressive external ophthalmoplegia, while one compound heterozygous patient had a complex HSP phenotype with progressive external ophthalmoplegia, ptosis, ataxia, moderate dysarthria, and proximal myopathy [37]. This phenotype is in accordance with the fact that the *paraplegin* encoded by *SPG7* gene is a mitochondrial protein. Our two HSP patients were homozygous for p.Leu78* and they shared a common haplotype between them and with the Italian patients originally reported by Arnoldi et al. [8], indicating that the mutation was introduced by a common ancestor. Both probands had disease onset in late adulthood; one had a postural hand tremor, while another had cervical dystonia and severe cortical and cerebellar atrophies. Thus, the p.Leu78*mutation is an ancient founder allele recurrently reported in patients of Roma origin, suggesting it might be a common cause of HSP in this ethnic group. The combined analysis of all patients carrying this mutation in the homozygous state (including the cases reported in this study) suggests late-onset HSP with common plus signs including ataxic, extrapyramidal, and myopathic signs. Further studies, especially in the European countries with large Roma communities, would contribute to exploring the prevalence and full spectrum of disease presentation of the p.L78* mutation that will have diagnostic applications.

One female patient had a de novo variant in the kinesin motor domain of the *KIF5A* gene causing SPG10. This form accounts for 2–10% of HSP cases in Europe, and it can present as a pure or complicated phenotype [38,39] with additional features including cognitive impairment, parkinsonism, polyneuropathy, hand muscle atrophy, autonomic impairment, deafness, and retinitis pigmentosa [10]. The age of onset in our patient was in late childhood, which has been previously reported, although the onset of SPG10 is usually after 35 years [17]. Our patient had complicated HSP with axonal polyneuropathy and mild postural hand tremors.

In one sporadic patient, a compound heterozygous mutation in the *ZFYVE26* gene causing SPG15 was identified. According to the literature, SPG15 is the second most common autosomal recessive HSP form and the second most common cause of HSP with thin corpus callosum after SPG11 [10,16,17]. Disease onset in our patients was in their teenage years, and the phenotype was complicated with thin corpus callosum, foot dystonia, and mild cerebellar signs (dysarthria, limb ataxia, and tremors). SPG15 resembles the phenotype of SPG11 with possible thin corpus callosum, intellectual disability, cerebellar ataxia, seizures, retinal involvement, polyneuropathy, and amyotrophy [10].

Our genetic analysis pipeline included genes that were reported to be more prominent at the time of design. Although our panel is still largely relevant, reflecting the more common causes of HSP (*SPAST, ATL1, REEP1, CYP7B1, SPG7,* and *SPG11*), it lacks genes like *KIF1A* (SPG30), which is currently known to cause 10–15% of all autosomal dominant HSP cases [40]. We acknowledge that a more comprehensive gene panel analysis would have improved our mutation detection. Still, this relatively small and easy-to-employ targeted-sequencing approach yielded a comparable diagnostic rate to larger studies and could be used as a first-line strategy for HSP molecular diagnosis. A recent cohort screening of 1550 HSP patients using a panel of 65 HSP genes achieved a diagnostic rate of 30.7% [7]. The most common genes in the study were *SPAST, SPG7, KIF1A, ATL1, SPG11, KIF5A,* and *REEP1*, which, except for *KIF1A*, are all included in our analysis, underscoring the relevance of our findings. In the Serbian HSP patient remaining with unknown genetic defects, further whole-exome or whole-genome sequencing should be performed to reach a higher diagnostic yield.

## 5. Conclusions

The combined genetic diagnostic yield of our gene panel and CNV analysis for HSP in the Serbian cohort was around 30%. The molecular diagnosis rate was higher in autosomal dominant compared to autosomal recessive and sporadic cases. The most commonly mutated genes were *SPAST* and *SPG11*. We also identified two patients of Roma origin with a homozygous founder mutation in the *SPG7* gene. Autosomal dominant patients usually had a pure phenotype while autosomal recessive cases usually had complicated HSP, including involvement of the cerebellum, extrapyramidal system, and peripheral nerves. Our results expand the genetic epidemiology of HSP. This study provides an overview of the genetic landscape of the HSP population in Serbia and can serve as a reference for establishing genetic diagnostic strategies in populations with a similar genetic background.

## Figures and Tables

**Figure 1 cells-11-02804-f001:**
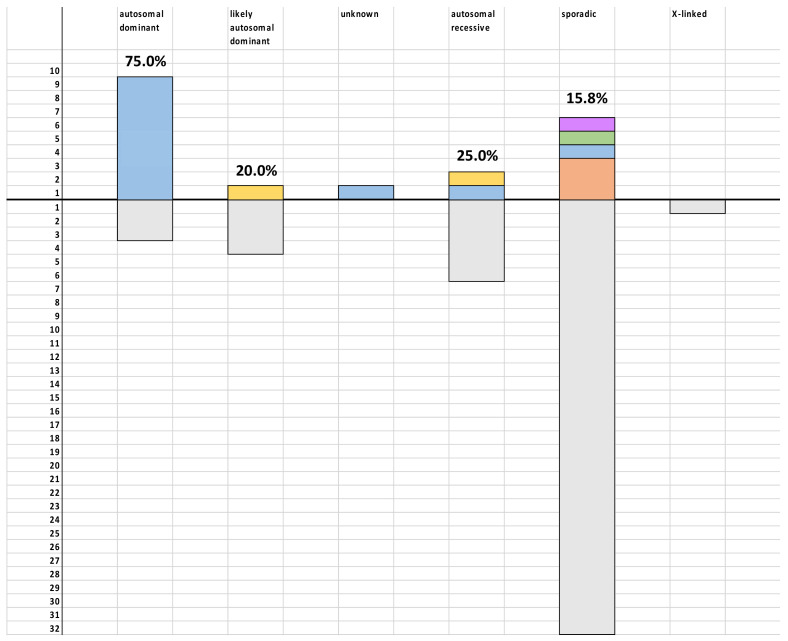
Percentage of HSP families with conclusive genetic findings based on the mode of inheritance. Legend: *X* axis—mode of inheritance, *Y* axis—number of families. Genetically positive patients are shown above the *X* axis, and genetically negative below *X* axis. *SPAST* gene mutations are given in blue, *SPG7* mutations in yellow, *SPG11* in orange, *KIF5A* in green, and *ZYFVE26* in violet. Percentage refers to percentage of patients with conclusive genetic findings among those with certain clinically defined mode of inheritance.

**Table 1 cells-11-02804-t001:** Genetic findings in the analyzed Serbian adult HSP patients.

Family ID	Patient ID	Presumed Inheritance	Disease Subtype	Gene Name	Transcript	Variant	Zygosity	HGMD Identifier #	gnomAD Frequency	ACMG Pathogenicity Class	ACMG Pathogenicity Evidence
1	HSP24	AD	SPG4	SPAST	NM_014946.3	c.1245+1delG	het	-	0	Likely pathogenic (novel)	PM2, PM5, PP1, PP3, PP4
	HSP94	AD	SPG4	SPAST	NM_014946.3	c.1245+1delG	het				
2	HSP32	AD	SPG4	SPAST	NM_014946.3	deletion of exon 15 and 16	het	CG1415382	-	Pathogenic (recurrent)	
3	HSP33	AD	SPG4	SPAST	NM_014946.3	c.425delA (p.Lys142Argfs*19)	het	-	0	Pathogenic (novel)	PVS1, PM2, PM4, PP1
4	HSP42	AD	SPG4	SPAST	NM_014946.3	c.1069_1070insA (p.I357Nfs*10)	het	-	0	Pathogenic (novel)	PVS1, PM2, PM4, PP1
5	HSP4	AR	SPG4	SPAST	NM_014946.3	c.1351A>G (p.R451G)	het	-	0	Likely pathogenic (novel)	PVS1, PM2, PM4, PP1
6	HSP68	sporadic	SPG4	SPAST	NM_014946.3	deletion exons 8–17 and further downstream	het	CG1415377	-	Pathogenic (recurrent)	
7	HSP52	AD	SPG4	SPAST	NM_014946.3	c.820_827del (p.M274Wfs*14)	het	-	0	Pathogenic (novel)	PVS1, PM2, PM4, PP1
8	HSP80	AD	SPG4	SPAST	NM_014946.3	c.1494G>T (p.R498S)	het	CM1512265	0	Pathogenic (recurrent)	
	HSP81	AD	SPG4	SPAST	NM_014946.3	c.1494G>T (p.R498S)	het				
9	HSP114	unknown (adopted)	SPG4	SPAST	NM_014946.3	c.308C>A (p.S103*)	het	-	0	Pathogenic (novel)	PVS1, PM2, PM4, PP1
10	HSP9	AD	SPG4	SPAST	NM_014946.3	c.1495C>T (p.R499C)	het	CM992676	0	Pathogenic (recurrent)	
	HSP10	AD	SPG4	SPAST	NM_014946.3	c.1495C>T (p.R499C)	het				
11	HSP11	AD	SPG4	SPAST	NM_014946.3	c.1672_1673del (p.L558Gfs*18)	het	CD021858	0	Pathogenic (recurrent)	
	HSP12	AD	SPG4	SPAST	NM_014946.3	c.1672_1673del (p.L558Gfs*18)	het				
12	HSP112	AD	SPG4	SPAST	NM_014946.3	deletion of 5’UTR-ex1	het	CG052756	-	Pathogenic (recurrent)	
13	HSP118	AR	SPG7	SPG7	NM_003119.3	c.233T>A (p.L78*)	hom	CM081826	0	Pathogenic (recurrent)	
14	HSP119	likely AD	SPG7	SPG7	NM_003119.3	c.233T>A (p.L78*)	hom	CM081826	0	Pathogenic (recurrent)	
15	HSP113	sporadic	SPG11	SPG11	NM_025137.4	c.5381T>C (p.L1794P)	hom	CM166061	0	Pathogenic (recurrent)	
16	HSP128	sporadic	SPG11	SPG11	NM_025137.4	c.5381T>C (p.L1794P)	comp het	CM166061	0	Pathogenic (recurrent)	
						duplication spanning intron27-ex29		CN166911	0	Pathogenic (recurrent)	
17	HSP35	sporadic	SPG11	SPG11	NM_025137.4	duplication spanning intron27-ex29	hom	CN166911	-	Pathogenic (recurrent)	
18	HSP34_HSP69	sporadic	SPG10	KIF5A	NM_004984.4	c.746T>C (p.Leu249Pro)	het	-	0	Likely pathogenic (novel)	PM2, PM5, PP1, PP3, PP4
19	DIST690	sporadic	SPG15	ZFYVE26	NM_015346.4	c.2114dupC (p.E706*)	comp het	-	0	Pathogenic (novel)	PVS1, PM2, PM4, PP1, PP3, PP4, PP5
						c.2357delC (p.P786Hfs*10)		-	0	Pathogenic (novel)	PVS1, PM2, PM4, PP1, PP3, PP4

AD—autosomal dominant, AR—autosomal recessive, het—heterozygote, hom—homozygote; novel mutations are highlighted; **#**—the breakpoints of all large-scale deletions/duplications are not characterized in detail. HGMD variant accession numbers for duplication/deletion variants encompassing the same exons as observed in this study are provided.

**Table 2 cells-11-02804-t002:** Main clinical findings of Serbian patients with genetically confirmed SPG4.

Features	Number of Patients
**% females**	50%
**Age at onset**	<1–6.2% 6–10–18.8% 11–20–6.2% 21–40–31.2% 41–60–31.2% >60–6.2%
**Age**	42.9 ± 11.
**Mobility**	normal—6.2% abnormal, but no aids—93.8%
**Lower limbs**	hyperreflexia—31.2% hyperreflexia, weakness—56.2% hyperreflexia, weakness, distal muscle atrophy—12.5%
**Upper limbs**	normal—18.8% hyperreflexia—75.0% hyperreflexia, distal muscle weakness—6.2%
**Sphincters**	normal—50.0% bladder impairment—12.5% bladder and bowel impairment—25.0%
**Sensibility**	normal—56.2% vibration impaired in lower limbs—37.5% vibration and touch impaired in lower limbs—6.2%
**Additional features**	56.2%

**Table 3 cells-11-02804-t003:** Clinical findings of Serbian patients with genetically confirmed SPG11.

Patient ID	HSP113	HSP128	HSP35
**Gender**	female	male	male
**Age at onset**	41–60	41–60	21–40
**Age**	52	53	25
**Mobility**	abnormal, but no aids	abnormal, but no aids	abnormal, but no aids
**LL**	hyperreflexia, weakness	asymmetric hyperreflexia	hyperreflexia
**UL**	normal	normal	hyperreflexia
**Sphincter dysfunction**	no	no	no
**Sensibility impairment**	normal	normal	normal
**Additional features**	foot dystonia	no	mild mental retardation (IQ 78), mild dysarthria, mild postural tremor
**NCS**	normal	bilateral carpal tunnel syndrome	normal
**SSEP**	not done	not done	abnormal
**Brain MRI**	normal	temporal arachnoid cyst	thin corpus callosum, mild periventricular WMHLs
**Spine MRI**	normal	normal	normal

LL—lower limbs, UL—upper limbs, NCS—nerve conduction studies, SSEP—somatosensory evoked potentials, MRI—muscle resonance imaging, WMHL—white matter hyperintensity lesions, IQ—intelligence quotient.

**Table 4 cells-11-02804-t004:** Haplotype sharing analysis for the p.L78* mutation in *SPG7*.

		Serbian p.L78* Patients				Italian p.L78* Patients			
		HSP118		HSP119		P1-EM9-06		P2-EM18-08	
Marker	Position (Mb)								
D16S3123	87.6500	99	101	101	109	101	109	101	109
rs8191483	88.8770	C	C	C	C	C	C	C	C
D16S3026	89.4930	202	202	202	202	202	202	202	202
D16S3121	89.4985	73	73	73	73	73	73	73	73
p.L78*	89.5769	A	A	A	A	A	A	A	A
rs12960	89.6203	G	G	G	G	G	G	G	G
rs174035	89.6277	C	C	C	C	C	C	C	C
rs455527	89.6440	T	T	T	T	T	T	T	T
rs352935	89.6486	C	C	C	C	C	C	C	C
rs2162943	89.7607	C	C	C	C	C	C	C	C

**Table 5 cells-11-02804-t005:** Clinical findings of Serbian patients with genetically confirmed SPG7.

Patient ID	HSP118	HSP119
**Gender**	male	male
**Age at onset**	41–60	41–60
**Age**	54	57
**Mobility**	unilateral support	abnormal, but no aids
**LL**	hyperreflexia, weakness	hyperreflexia, weakness
**UL**	hyperreflexia	hyperreflexia
**Sphincter dysfunction**	no	bladder
**Sensibility impairment**	no	vibration in LL
**Additional features**	postural tremor	torticollis
**NCS**	not done	normal
**SSEP**	not done	not done
**Brain MRI**	one small supratentorial WMHLs	severe cortical and cerebellar atrophy
**Spine MRI**	normal	normal

LL—lower limbs, UL—upper limbs, NCS—nerve conduction studies, SSEP—somatosensory evoked potentials, MRI—muscle resonance imaging, WMHL—white matter hyperintensity lesions.

**Table 6 cells-11-02804-t006:** Clinical findings in patients with genetically confirmed SPG10 and SPG15.

Patient ID	HSP34	DIST690
**HSP type**	SPG10	SPG15
**Gender**	female	female
**Age at onset**	6–10	11–20
**Age**	20	23
**Mobility**	abnormal, but no aids	abnormal, but no aids
**LL**	hyperreflexia, weakness	hyperreflexia, weakness
**UL**	hyperreflexia	normal
**Sphincter dysfunction**	bladder	no
**Sensibility impairment**	vibration and touch in LL	normal
**Additional features**	foot deformities, mild static hand tremor	foot dystonia, mild dysarthria, mild limb ataxia and tremor
**NCS**	predominantly motor, axonal polyneuropathy	normal
**SSEP**	abnormal	normal
**Brain MRI**	normal	normal
**Spine MRI**	normal	normal

LL—lower limbs, UL—upper limbs, NCS—nerve conduction studies, SSEP—somatosensory evoked potentials, MRI—muscle resonance imaging.

## Data Availability

All obtained data are presented within the body of manuscript, figure and tables.

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
