# Peer review of "Phenotypic and Genetic Heterogeneity of Adult Patients with Hereditary Spastic Paraplegia from Serbia"

_cells, 2022, doi:10.3390/cells11182804_

Round 1
Reviewer 1 Report
The manuscript by Peric et al reports a clinical and genetic study of a large “homogeneous” series of patients from 65 families with adult onset hereditary spastic paraplegia from Serbia. Using an amplicon-based next generation sequencing and MLPA analysis focused on 13 genes involved in these diseases, they were able to detect pathogenic mutations in 30% of the patients, some are novel.
The manuscript is well written and clear but may be improved as follows:
- The rational for the selection of 13 genes while 80 are involved is not indicated. Are they those involved in adult forms mainly or the more frequent? Some are rarely affected such as SPG78 or SPG15 or BICD2. The same for the CNV analysis that was restricted to 3 genes only. A recent study from Mereaux et al (2022) showed that the majority of mutations are found in a relatively limited number of genes but compared to this study, some are lacking here while found frequently in the published series.
- The description of the full cohort is lacking. Only the fact that they are adult forms is indicated but the readers will appreciate to have access to the mean and range of the age at onset and examination, and men / women ratio for example. How many patient per family were analyzed? What about sporadic forms, were trio sequencing performed? Was segregation studies performed for the novel variants?
- In the introduction, there is the lack of a more recent review on spastic paraplegias.
- The late onset of SPG11 cases is unusual compared to most published cases and could be discussed, particularly regarding the presence of missense variants also reported in Italian cases with late onset by Rubegni et al (2015).
- Table 2 may be simplified with only the frequencies of each signs and the full table put on supplemental data.
- Regarding the sporadic SPG11 cases, were the parents available to test if de novo?
- It would have been interesting to compare the haplotypes of the SPG7 cases with the same mutation before stating that they are from the same founder as stated in the conclusion section. The pedigrees of the 2 families may also be informative in regards to the multiple cases with possible pseudo-dominance indicated in the results section.
- The diagnosis of 30% of the cases is very similar to other studies, including studies with much more genes tested. This may be discussed.
- There is no discussion regarding the strategy proposed for the negative cases (deep-intronic analysis, exome etc…).
Reviewer 2 Report
The manuscript describes comprehensive mutational analysis focusing of L1CAM (SPG1), PLP1 (SPG2), ATL1 (SPG3A), SPAST (SPG4), CYP7B1 (SPG5A), SPG7 (SPG7), KIF5A (SPG10), SPG11 (SPG11), ZYFVE26 (SPG15), REEP1 (SPG31), ATP13A2 (SPG78), DYNC1H1, and BICD2, using next generation sequencing-based techniques. Sanger sequencing was additionally used to obtain nucleotide sequences that were not covered by nest generation sequencing. CNV analysis based on MLPA analysis was conducted for SPAST, SPG7 and SPG11.
The mutational analysis was comprehensive and the obtained results were convincing. The reviewer raises the following comments:
1. Although the authors described that interpretation of variants were based on the American College of Medical Genetic and Genomics (ACMG) guidelines, details were not provided. In particular, interpretation of novel likely pathogenic variants needs to be fully explained. Allele frequencies in the populations should also be provided.
2. Detailed description of cognitive functions in HSP113, HSP128 and HSP135 should be provided. It is unclear what was meant by “corpus callosum dysgenesis”. How different is “corpus callosum dysgenesis” from thin corpus callosum?
3. The first paragraph of Discussion is largely overlapped with those described in Results, and could be deleted.
4. In the first paragraph of page 4 (Discussion), “internal exon sequences” needs to be clearly explained.
5. Although the method employed in this study is well established, the present study analyzed a limited number of genes. The authors are encouraged to discuss on limitations of the methods used in this study. Comprehensive sequencing (for example, exome sequencing) would be expected to produce higher diagnostic yields.
